# Personalized Biometrics of Physical Pain Agree with Psychophysics by Participants with Sensory over Responsivity

**DOI:** 10.3390/jpm11020093

**Published:** 2021-02-02

**Authors:** Jihye Ryu, Tami Bar-Shalita, Yelena Granovsky, Irit Weissman-Fogel, Elizabeth B. Torres

**Affiliations:** 1Rutgers University Center for Cognitive Science, Psychology Department, Rutgers University, Piscataway, NJ 08854, USA; jihyeryu@mednet.ucla.edu; 2Department of Occupational Therapy, School of Health Professions, Faculty of Medicine, Sagol School of Neuroscience, Tel Aviv University, Tel Aviv 6997801, Israel; tbshalita@post.tau.ac.il; 3Laboratory of Clinical Neurophysiology, Department of Neurology, Faculty of Medicine, Rambam Health Care Campus, Technion, Haifa 3109601, Israel; y_granovsky@rambam.health.gov.il; 4Physical Therapy Department, Faculty of Social Welfare and Health Sciences, University of Haifa, Haifa 3498838, Israel; ifogel@univ.haifa.ac.il; 5Center for Biomedicine Imaging and Modeling, Computer Science Department, Rutgers University, Piscataway, NJ 08854, USA

**Keywords:** EEG, pain biometrics, stochastic analyses, micro-movements spikes, sensory over responsivity, standardized scale, personalized pain

## Abstract

The study of pain requires a balance between subjective methods that rely on self-reports and complementary objective biometrics that ascertain physical signals associated with subjective accounts. There are at present no objective scales that enable the personalized assessment of pain, as most work involving electrophysiology rely on summary statistics from a priori theoretical population assumptions. Along these lines, recent work has provided evidence of differences in pain sensations between participants with Sensory Over Responsivity (SOR) and controls. While these analyses are useful to understand pain across groups, there remains a need to quantify individual differences more precisely in a personalized manner. Here we offer new methods to characterize pain using the moment-by-moment standardized fluctuations in EEG brain activity centrally reflecting the person’s experiencing temperature-based stimulation at the periphery. This type of gross data is often disregarded as noise, yet here we show its utility to characterize the lingering sensation of discomfort raising to the level of pain, individually, for each participant. We show fundamental differences between the SOR group in relation to controls and provide an objective account of pain congruent with the subjective self-reported data. This offers the potential to build a standardized scale useful to profile pain levels in a personalized manner across the general population.

## 1. Introduction

The peripheral nervous systems include an interconnected network of afferent nerve fibers carrying information from the skin to the spinal cord and onto the brain [1]. This flow of activity can be modeled as it updates the brain moment by moment, reflecting the trajectories of our bodies in motion [2,3] or of the fluctuations in bodily signals at rest [4,5,6,7], within a given environment where sensory input is processed and integrated with ongoing movements making up intended [8,9] or spontaneous [10] behavioral states. The afferent fibers from the periphery carry information about touch, pressure and movements sensed by the mechanoreceptors [11], while thermoreceptors and nociceptors process information about temperature and pain, respectively [1,12]. Collectively, they give rise to the sense of touch, which is important to manipulate objects [13], to control our movements [14], to gain a sense of body ownership [15] and affection [16], and to develop and maintain our overall psychological and social wellbeing [17].

The experience of pain (i.e., its subjective perception) is comprised of sensory, affective-emotive, and cognitive processes of a noxious input. Pain experience can be measured in the lab applying quantitative sensory testing, namely inducing measurable pain stimuli of different modalities (e.g., heat, pressure), while subjects are required to rate their pain intensity/unpleasantness using various pain scales (e.g., visual analog scale, numerical rating scale). Thus, the individual’s experience of pain though seemingly centrally processed, it is evoked at the periphery using different experimental assays. These may include (among others) the physical experience of sustained pressure [18,19] or sustained temperature [20,21,22,23], carried along peripheral afferent nerves to the central nervous systems, which is comprised of the spinal cord and the brain.

In recent years, we have learned about the central processing of movement-related reafference from a special participant (Ian Waterman, IW) who experienced a viral infection that killed the afferent fibers for light-touch, pressure, and movements. The infection spared the afferent fibers for pain and temperature [24,25,26]. IW has remastered motor control in the absence of proprioception and kinesthetic reafferent information, by sensory substituting with vision the senses of touch, pressure, and movement [27,28]. Perhaps using information about his central processing of peripheral activity during resting state [24], could help us develop new models of statistical inference and interpretations for use in other data sets. His case could help us interpret resting-state data from centrally processed sensory information in other patient populations with sensory processing dysfunctions mediated by disruptions in peripheral reafferent flow [29,30].

Ian Waterman’s case is interesting as fluctuations in his electroencephalographic (EEG) activity at rest revealed the presence of the exponential distribution of peak amplitudes (Figure 1). This distribution represents a memoryless random process whereby past activity does not contribute to the probabilistic prediction of future events. In this case, events refer to moment-by-moment fluctuations in signals’ amplitudes and timings. We posit that these fluctuations inform the nervous systems of dynamically adaptive states, as they transition from highly variable to steady-state. Based on our prior theoretical work on kinesthetic reafference [8], we have conjectured that this type of memoryless process may impede creating a proper memory buffer to sustain activity long enough to bring it to the brain’s awareness, to consciously recognize it, or to use it effectively as reference to inform and predict impending states of the system [5,8,29].

Having found in IW these patterns at rest, reflecting the variability of the signals as a renewal process in “the here and now” in the absence of movement reafference sensations, may help us characterize other states related to pain sensation in neurotypical controls. More precisely, it may also help us characterize, stochastically, the departure from this memoryless state, in cases with atypical pain sensations. We know the stochastic signatures of not sensing touch, pressure, and movement, in a person that nevertheless senses temperature and pain. As such, we may use this prior information as reference to learn how the fluctuations in EEG activity may distribute during resting state for a person who does not have severed communication between the peripheral afferent fibers and the brain, but that nevertheless reports atypical sensation of pain. We would like to assess distributions of stochastic activities related to fluctuations in EEG peak-amplitudes on participants with sensory over-responsivity (SOR), a subtype of sensory modulation dysfunction (SMD) which in turn falls under the broader umbrella of sensory processing disorder [31].

The SOR subtype of SMD manifests clinically as a condition in which stimuli that are not typically painful are perceived as abnormally irritating, unpleasant, or even reportedly painful [32], sometimes interfering with activities of daily life [33]—as measured by several clinical scales. These clinical manifestations are also consistent in laboratory experiments, measured under controlled conditions [34]. Under these controlled settings, people with SOR express discomfort and hypersensitivity to experimental manipulations in pressure or temperature, whereby the lingering sensation of evoked peripheral activity leads to the conscious expression of pain and sustained pain aftersensation centrally experienced [34].

Prior work has relied on population statistics and provided an account of full cohorts. A new detailed individualized characterization of minute fluctuations in EEG activities while experiencing pain could help us re-examine these issues to formulate a personalized account, useful to inform automatic groupings and stratifications of random draws of the population, with the overarching aim of defining a standardized scale of centrally processed pain. This would be beneficial to other disorders on a spectrum (e.g., autism, schizophrenia, and Parkinson’s disease) whereby such sensory processing issues of pain abound too [35,36,37,38]. Across these various disorders of the nervous system, we need proper objective characterizations of pain sensation to complement and augment reports on the subjective sensations of pain captured by clinical inventories.

The type of analysis that we offer here, away from assumptions of theoretical population statistics, has been previously used on a characterization of stochastic variations in movement reafferent signals. This is a data-driven approach whereby we let the data reveal patterns and then, upon interpretation and inference, we propose possible lines of inquiry to pursue in future work. In our prior work, the results led to automatic clustering of the above-mentioned clinical disorders on a spectrum [39]. These in turn, have shown strong ties with other disorders of the nervous systems and various types of disruption in reafferent flow of movement information [40,41], thus allowing us to further pursue new lines of questions. Since pain and temperature share separable afferent channels from movement afference, and crosstalk can be quantified through central processing using controlled experimental assays, here we apply these new data-driven analytical methods to SOR participants who suffer from abnormally high pain sensation. We re-examine previously published EEG data [34] as well as explore pain-evoked EEG responses induced by sustained temperature in controls vs. SOR participants. We do so by analyzing the gross data commonly discarded as noise, by avoiding a priori assumptions of theoretical normal distributions of the fluctuations in EEG-waveforms’ peak amplitudes. We discuss our results within the context of stochastic processes amenable to offer a probabilistic account of pain sensation in general.

## 2. Materials and Methods

These details of the experiment have been explained in previous publications, but we report them here for completeness [34,42]. The Rutgers University Institutional Review Board approved this de-identified data sharing. The IRB committee of Rambam Health Care Campus approved this study in 2013. The IRB number is 3075, The Israeli ministry of health # HT4858.

### 2.1. Participants

The study included 21 healthy participants (5 males and 16 females) between the ages 18 and 40 years old recruited from a convenience sample in a laboratory database. Participants were naïve to the testing. Based on a medical survey, participants with no chronic pain history and no regular use of analgesic or psychiatric medication were included in the study. Participants with any psychological, psychosocial, metabolic, and neurological disorders were excluded from the study. This means that if the participant had a diagnosis of any of the above-mentioned disorders, they did not sign up, nor did they participate in the study.

Participants were able to communicate and understand the instructions of the study. They self-reported to be free of any pain relief medications 24 h prior and any caffeine products at least 2 h prior to the experiment, and to have had sufficient sleep the night before. Sufficient sleep means that participants did not express any complaints about sleep disturbances. We did not measure their sleep. All participants provided written consent, which was approved by the Institutional Review Board of Rambam Health Care Campus (Haifa, Israel).

Participants were categorized into two groups—sensory over responsiveness (SOR) group (*n* = 9, 1 male) and control group (*n* = 11, 3 males). The SOR group was comprised of those whose Sensory Responsive Questionnaire Intensity Scale (SRQ-IS; [43])-Aversive score exceeded 2 standard deviations from its mean. The control group was comprised of those with scores within the 2 SD from the mean. Note, the SRQ-IS is designed to clinically identify those with sensory modulation disorder and is comprised of Hedonic scores and Aversive scores [44]. The Aversive scores that were used as a criterion to categorize groups involve answering intensity levels (on a scale 1–5) on scenarios such as “Being in dark/unlit surroundings bothers me,” and “Watching T.V./computer in a well-lit room bothers me.” Further details of participant recruitment can be found in [34]. By using these scores, we operationalized each participant’s perception of sensory experience.

### 2.2. Experiment

This was a block design experiment, whereby each participant performed all three conditions in blocks of trials. They sat comfortably in a quiet air-conditioned room under all 3 conditions. In the first condition, the participant was instructed to close his/her eyes and rest for 5 min. In the second and third condition, pain was administered for 5 min. Note, both conditions are identical and merely sequential in order. In each of these pain conditions, heat stimulus was applied to the participant’s forearm with 8–12 s interval to simulate a pain experience. Specifically, the participant was applied with a heat stimulus by the Contact Heat-Evoked Potential Stimulator, which is a computerized thermal stimulator (Medoc Ltd. Advanced Medical Systems, Ramat Yishai, Israel). The temperature was tailored to everyone to evoke a peak pain magnitude of 50/100 (pain-50) on the numeric rating scale. Specifically, we gave 30 stimuli, ISI 8-10, baseline temperature 39 °C with destination pain 50 described in [42] +0.5 °C. After each stimulus, during ISI, subjects provided pain intensity and pain unpleasantness ratings, using the numerical rating scale. During the study, EEG signals were recorded with a 32-electrode cap (Easy Cap Q40; FMS Falk Minow Services, Herrsching, Germany) with the Quick Amp EEG System (Brain Products GmbH, Munich, Germany). These signals were processed at 500 Hz sampling rate, with 0.15–100 Hz bandpass filter, and a notch filter at 50 Hz. The EEG signals were further preprocessed using the PrepPipeline toolbox [45], with which we referenced via a robust average reference procedure, where channels were iteratively referenced to the average signal, while bad channels, such as those showing extreme amplitudes (deviation z-score exceeds 5) or lacked correlation with any other channel (correlation less than 0.4), were excluded and interpolated in this process.

### 2.3. Data Analysis

#### 2.3.1. Analyses in the Frequency Domain

For each condition, pairwise cross-coherence was computed using each of the 32 EEG channel waveforms (Figure 2A). Across the frequency range within the cross-coherence values, we extracted the maximal value within the beta and gamma bands (13–100 Hz), as this bandwidth showed to have a noticeable difference between the SOR and control groups (Figure 2B). Note, we had examined other bandwidths, as well as beta and gamma band separately, but did not find such a pattern. For that reason, we focused on the beta and gamma bands combined.

From the maximal cross-coherence values obtained, we built an adjacency matrix for each participant from each condition (Figure 2C). Based on this matrix, first, we categorized the channels by scalp areas—frontal (F), temporal (T), parietal (P), and occipital—and compared the median of maximal cross-coherence values between different combinations of scalp areas (Figure 2D).

Further, using the adjacency matrix, we built a network where the nodes correspond to a single EEG channel’s activity, and edge corresponds to the maximal cross-coherence value between the two nodes. Here, network edges between a set of nodes form triangles, and the fraction of triangle numbers formed around each node is defined as the cluster coefficient. This is a measure of segregation within a network and is computed using the average intensity (geometric mean) of all triangles associated with each node using an algorithm by [46]. Equation (1) describes the computation, where N is the set of all nodes, *C_i_* is the Cluster Coefficient of node *i* (out of *n* = 32 nodes); *t_i_* is the geometric mean of triangle links formed around node *i* and *k_i_* is the number of degrees (links) formed around node *i*. The median of these cluster coefficients from all EEG channel was then computed for each participant and compared across different groups and conditions using the Kruskal–Wallis nonparametric test.
(1)$$C_{i}=\sum_{i\in N}\frac{t_{i}}{k_{i}\left(k_{i}-1\right)}$$

#### 2.3.2. Analyses in the Temporal Domain

Among the 32 EEG channels, we selected a channel with the highest cluster coefficient, as it would be deemed a hub channel, and analyzed its temporal data. The location of the selected channel can be found in Figure A1. Specifically, we bandpass filtered the data at 13–100 Hz using IIR filter at 20th order (Figure 2E). Then we extracted the micro-movement spikes (MMS).
(2)$$\mathrm{MMS}=\frac{local\;peak}{local\;peak+avg\left(activity_{\min to\min}\right)}$$

This standardization equation is commonly used to address allometric effects (Mosimann, 1970) that occur due to individual anatomical differences (Figure 2F).

Micro-movements spikes (MMS): To standardize the amplitudes of the data, we shifted the data up so that the minimum value of the waveform equals 0. Then, to compute a set of standardized spike amplitudes, we took each spike amplitudes from the filtered and shifted waveforms and divided this local peak by the sum of this raw spike amplitude value and the average of the signals sampled within the two adjacent minima surrounding that local spike, as shown in Equation (2).

To examine the change in stochastic variations of the signals over time, we extracted the MMS due to fluctuations in the signals’ amplitude from each condition (of 5 min duration) and examined how the frequency distribution of these standardized spike amplitudes changed over time. Specifically, we segmented the data by 4 s time window, while sliding it with 50% overlap between consecutive windows. This allowed us to gather on average 100 spikes per window (the criteria to have proper statistical power for our 95% confidence in the empirical estimation.) For each time window, histograms of MMS peaks’ amplitudes were plotted, binned from 0.5 to 0.7 with 0.02 intervals. Then, we used similarity metric that enables us to compare probability distributions pairwise and estimate differences in probability space. We obtained the earth mover’s distance (EMD) [47,48,49] between 2 sequential windows’ histograms to quantify the change in stochasticity (Figure 2G). The EMD (also known as the Kantarovich–Wasserstein distance [47,48,50,51]) is a distance metric that can quantify stochastic shifts in probability space. Previous work elaborates on the algorithm to compute this distance adapted to our biometrics [7]. The stochastic shifts in the EMD across the data set were thus examined, by obtaining the distribution of EMD values (Figure 2H) using Freedman–Diaconis binning rule [52] and fitting a Gamma probability distribution function (PDF) using maximum likelihood estimation with 95% confidence intervals (Figure 2I).

The Gamma PDF is defined by two parameters—the shape and the scale—and these parameters are informative to provide interpretations of stochastic features of a single participant by localizing the participant’s signatures (empirically estimated) on the Gamma parameter plane (Figure 2I.) Then we can interpret this personalized signature in relation to other participants localized under similar conditions, and in relation to the baseline signature of the participant as we vary the conditions (e.g., from resting state to pain, to de-adaptation from pain).

The continuous family of Gamma probability distribution functions (PDF) ranges from exponential (shape equals 1, representing the case of the memory-less exponential distribution) to skewed, asymmetrical distributions with heavy tails, to Gaussian-like symmetric distributions (with higher shape values). By sampling over large numbers of nervous systems biorhythms sampled from the human population, across disorders of the nervous system, ages, and between sex, we have empirically found a power law relating the shape and the scale parameters. In this empirically found relation, as the shape values increase, the scale values decrease consistently with a tight linear fit on the log-log Gamma parameters’ plane spanned by the values of the shape and scale. The scale values represent the noise to signal ratio, NSR (i.e., empirically estimated Gamma variance over the Gamma mean). Knowing one parameter (the shape) helps us infer the other (the scale), owing to this power-law relation. We have empirically found that processes with high noise (high scale value) and close to the random exponential distribution (small shape value) correspond to stochastic regimes of high uncertainty, leading to poor prediction of future events from present events. Likewise, processes with symmetric distributions (high shape values) and low NSR correspond to stochastic processes with high certainty, describing predictive performance with high accuracy. This has been the case for data related to central signals registered from EEG and resting-state fMRI processing, and for peripheral signals registering kinematics of different movement classes. This has also been the case for autonomic signals related to heart and breathing activities [53,54,55,56].

## 3. Results

### 3.1. SOR Participants Show Reduced Cortical Interactions within the Beta and Gamma Bands during Resting Condition

As a first step, between all pairs of channels, we obtained the cross-coherence measure and extracted the maximum for each comparison. We extracted this information within the beta and gamma bands (13–100 Hz). By categorizing the channel pairs by their corresponding scalp areas, we find an overall lower coherence among the SOR group than the control group. This is most noticeable from the interactions between temporal and frontal (χ(1,19) = 4.69, *p* = 0.03), parietal and temporal (χ(1,19) = 4.05, *p* = 0.04), and occipital and temporal areas (χ(1,19) = 4.37, *p* = 0.04) (Figure 3A). Such reduced coherence among SOR were observed only during the resting condition, and not during the pain induced conditions, during which the coherence levels were similar between the two groups. From this, we find that reduced cortical interactions within the beta and gamma bands during resting condition is characteristic of SOR.

As a subsequent analysis in the frequency domain, we used the adjacency matrix of pairwise cross-coherence values to create a network graph and quantify the connectivity across all channels. We computed the clustering coefficient value for each channel and obtained the median to compare values between the two groups, controls and SOR.

During the resting condition, SOR showed lower clustering coefficients than the control group (χ(1,18) = 4.37, *p* = 0.04), implying a more sparse connection across the scalp within the beta and gamma band. On the other hand, under the two pain conditions, the connectivity remained similar between the two groups (Figure 3B).

### 3.2. Relative to Baseline, SOR Participants Show Higher Rates of Change in Stochastic Signatures than Controls

The temporal stochasticity of the most connected channel (i.e., channel with highest cluster coefficient) was examined by band-passing the time series through the beta and gamma band (13–100 Hz), extracting the MMS amplitudes, and building a stochastic trajectory on the Gamma plane. We then examine the first-order rate of change in Gamma parameter position, using the frequency histogram of the MMS peaks and computing the EMD between two consecutive histograms (PDFs.) This amounts to a “speed temporal profile” of the PDFs as they shift stochastic signatures per unit time on the Gamma parameter plane. Our unit time is 4 s time window, enough time to make an empirical estimation with statistical power and high confidence (based on frequency histograms derived from over 100 peaks.) The EMD values thus obtained per two consecutive time windows were then accumulated into a frequency histogram, and the distribution of EMDs compared between the two groups for each condition.

As shown in Figure 4A, in the resting condition, the SOR group showed a more symmetrical distribution of EMD values, reflected by its higher shape (χ(1,19) = 8.91, *p* < 0.01), and lower scale fitted parameter values (χ(1,19) = 6.91, *p* < 0.01) than the control group. As shown by the symmetry of EMD distribution from the SOR group, the MMS amplitudes of the cortical signal tends to be more predictive with reduced noise. Conversely, the typical individual from the control group tends to have an exponential-like distribution of EMD values, implying that their signals tend towards a memoryless regime, where the past is not informative to predict the future.

In the first pain condition, when the pain was induced for the first time, the SOR group and the control group started to show less distinction in their stochasticity. The typical individual from the control group did not change too much from the resting condition, where the EMD values were distributed closer to an exponential (memoryless) distribution. However, in this condition, the SOR group started to exhibit a pattern similar to the control group, shown by a reduced shape value and higher scale parameter values. Nevertheless the distinction is still statistically significant between the two groups for the shape (χ(1,19) = 4.85, *p* = 0.03) and borderline significant for the scale (χ(1,19) = 3.41, *p* = 0.06) parameter.

In the second pain condition, the SOR group and control group no longer exhibited their distinction with statistical significance, as quantified by the shape (χ(1,19) = 0.61, *p* = 0.43) and scale (χ(1,19) = 0.61, *p* = 0.43) values. At this point, both groups show their EMD values to be distributed closer to an exponential distribution, implying that their MMS amplitudes of the cortical signals have higher uncertainty, with higher noise and randomness relative to baseline.

At a different angle, when we also examine the empirical moments of the EMD distribution between the two groups, we found some distinction in the resting condition. In general, the SOR group tended to have a higher mean (χ(1,19) = 2.91, *p* = 0.09) EMD values implying higher rates of stochastic change from their baseline state, compared to controls. Although other moments were not statistically different in their values (variance χ(1,19) = 0.01, *p* = 0.94; skewness χ(1,19) = 0.85, *p* = 0.35; kurtosis χ(1,19) = 0.50, *p* = 0.48), within each condition, the variance and skewness tend to have a tighter range across individuals for the SOR than the control group (Figure 4B).

Given the significant differences in the rates of stochastic shift between controls and SOR relative to baseline and their non-significant statistical difference during pain and pain recovery conditions, and given the result that controls do not shift from the exponential regime during pain conditions, we can safely conclude that the EEG beta and gamma bands of the EEG signals from the SOR experienced significantly higher shifts at a faster rate than controls did when transitioning from condition to condition. Their stochastic signatures localize these two groups on different probability distributions and the shifts in probability space are larger in magnitude and rate for SOR.

### 3.3. Inventory Scores Agree with Stochastic Characterization of Brain EEG Signals’ Fluctuations

With an aim to find correspondence between metrics obtained from different domains—temporal, frequency, and clinical inventory scores—we visualized these together as shown in Figure 5A for EMD fitted shape parameter, cluster coefficient (CC), and SRQ-IS score, and in Figure 5B for EMD fitted scale parameter, cluster coefficient, and SRQ-IS score. Overall, the EMD shape parameter has a strong relation with CC and the SRQ-IS scores, and thereby separate the SOR group from the control group well. On the other hand, EMD scale parameter has some relation to those metrics, but to a lesser degree.

To examine the correspondence to a finer level (Figure 5C), we median ranked the participants by quartiles along the shape and scale parameters and along the CC values, and compared the quartile groups’ inventory scores, CC, and Gamma parameter values using a non-parametric Kruskal–Wallis test. Specifically, we compared between the 1st and 2nd quartiles, between 3rd and 4th quartiles, and between the lower 50 percentile and the higher 50 percentile (i.e., the 1st and 2nd quartiles combined against the 3rd and 4th quartiles combined). In general, when comparing the lower and upper 50 percentiles, the Gamma shape parameter had a strong correspondence with CC (χ(1,19) = 6.43, *p* < 0.01) and the inventory score (χ(1,18) = 4.55, *p* = 0.03); and the Gamma scale parameter exhibited such relation, but to a lesser degree with CC (χ(1,19) = 3.61, *p* = 0.06) and inventory score (χ(1,18) = 3.45, *p* = 0.06). However, for both Gamma parameters, their statistical significance was only observed when comparing the lower and upper 50 percentile, which is roughly the separation of the SOR group against the control group. When we examine at a finer level, to compare within the SOR group and within the control group, such correspondence is hard to see for all 3 metrics.

## 4. Discussion

This work aimed at offering a new characterization of central signals from EEG activities registered during baseline state, and pain conditions in participants with SOR, relative to controls. We successfully reproduced previously published results including population-based statistical analyses in [34] whereby the baseline EEG activities of SOR during resting state significantly differed from controls. Further, we add new findings to the objective characterization of pain.

In the present analyses, we employed a personalized approach whereby we made use of the gross data (i.e., all fluctuations away from the empirically estimated mean of the person’s data) that is usually discarded as noise. We characterized each participant’s gross data by the MMS of EEG signals’ amplitude, and empirically estimated the continuous family of probability distributions that best fitted these fluctuations for each participant in an MLE sense. We then uniquely localized each participant on a probability parameter space. Using this information, and a proper distance metric to measure change in probability space, we then tracked for each participant and for the entire cohort, the rates of change in stochastic shifts, when transitioning from resting state to pain 1 and to pain 2 conditions.

This individualized characterization of the brain EEG activity revealed two fundamental differences between SOR and control participants: (1) the distributions of the EMD signaling stochastic shifts was exponential in controls and tending to symmetric in SOR; (2) the shifts in the shape of this type of probability distribution in controls was not visible (i.e., they remained exponential) during the pain conditions, but significantly shifted from more to less symmetric shapes, to exponential, in SOR participants. Lastly, we found good correspondence between the clinical classification scores and the stochastic signatures that we empirically estimated for each participant, signaling that our personalized approach is not at odds with the clinical approach. This is important to augment the subjective inventories reflecting the person’s self-perception of pain, with the objective biometrics quantifying the physical sensations of pain evoked by this experimental assay. The type of temperature-based manipulation used by the assay occurs at the periphery. Through afferent flow, the processing, transduction, and transmission of these signals from the peripheral to the central nervous systems give rise to physiological EEG signals reflecting the brain activities during these conditions. When the fluctuations in these probability distribution signatures are exponentially distributed, random memoryless, and with high NSR, the peripheral stimuli are not perceived as painful (controls cases). When the shifts in signals are distributed with quasi-symmetric shapes tending to the Gaussian distribution, the stimuli are perceived as a lingering sensation of discomfort and reported as pain (SOR cases.) As such, our work here offers a set of biometrics whereby the perception of pain levels coincides with the physiological (physical) sensation of bodily signals. Peripheral changes are centrally registered both at the level of the stochastic shifts in EEG signals and at the level that the person can consciously self-report.

The discovery that the baseline signatures in controls are exponential when the SOR signatures are Gaussian-like lends itself to the following interpretation (in light of what we know from reafferent signals in the resting state EEG activities of deafferented participant IW, who cannot sense movement): The controls’ baseline activity with a random, memoryless regime that does not change much during pain conditions, implies that there is not enough buffering of the activity to sustain the sensory information and use it as an anchor to predict impending events (signal’s fluctuations) in the pain condition. The control participant experiences the baseline and the pain in “the here and now” with a renewal process that is too random and variable (with high NSR) to systematically sustain a memory of the events and anticipate impending spiking activity in the context of pain. As such, the control participant does not reportedly sense pain, because this information does not shift stochastic signatures from baseline and at baseline, the information is just random background noise. In stark contrast, the SOR participant starts out at resting state with systematic signals that have higher shape values (more symmetric distributions) and lower scale values (lower NSR) implying higher statistical certainty. This higher certainty is amenable to build a more reliable predictive code whereby impending variations in the signals can be systematically anticipated, thus scaffolding the ability to build a memory buffer to consciously register the change from resting to pain state. In this sense, the physical pain at the periphery surfaces to consciousness as the brain activity seems to offer more awareness of change in SOR than in the controls’ signals, which remain as random noise.

When transitioning from Pain1 ➔ Pain2 condition, the data from SOR participants shows a trend that approaches the controls. We interpret this as an adaptive phenomenon. As the system adapts to the lingering sensation of pain, the Pain1 ➔ Pain 2 case, the SOR activity returns to the exponential (random and memoryless) case whereby the person does not feel the lingering sensation of pain with the same intensity as it did in the Resting ➔ Pain 1 condition. The activity seemingly went back to a random memoryless state with no memory (no buffering of the activity long enough to bring it up to conscious perception) thus not sustaining the lingering sensation with the same intensity as in the initial block of the experiment. In this sense, the proposed stochastic-process interpretation of the pain sensation is to have these two opposing limiting states along a continuum (random memoryless vs. predictive) instantiated by the distribution of the signals and how they change from moment to moment. The EMD in this case provides information about the shifts of the frequency histograms representing probabilities derived from the signals’ fluctuations. Of course, this is merely a proposition and will need validation with larger N, but we express this caveat in the section below, referring to these issues.

Our results of treating everyone (individually) as a random process and empirically characterizing the individual stochastic signatures and their rates of change during pain states, invites a new characterization of pain states in relation to resting states. This personalized characterization is also amenable to examine the cohort behavior and identify statistical self-groupings congruent with clinical scores. We see that the physical sensation of pain is perceived and reported by the person with SOR but not by the control participant, whose activities do not sustain, nor anticipate the pain state.

In summary, the changes in EEG MMS that we quantified in the beta and gamma band (13–100 Hz) may reflect the renewal processes in central neural processing that is continuously refreshed by the peripheral feedback from afferent signals. The activity of the gamma band alone (putatively related to attentional states) or of the beta band alone (putatively related to movement afference) will not produce these patterns. It is their combined activities that brings the signal that reflects their integration as the person reaches awareness of the lingering sensation of pain (Resting ➔ P1), or as the system deadapts from it in the second block (P1 ➔ P2).

A predictive shift in the MMS of this combined signal, as quantified by the EMD distribution, may imply that participants with SOR perceive a lingering effect of the painful experience. We propose it as a systematic predictive memory of it that is sustained long enough to bring it to awareness. This may be through increased certainty experienced by increasingly systematic prediction-confirmation loops, away from randomness. This interpretation, which is further supported by the congruence of our statistical inference with the clinical scores, warrants further investigation, given the critical need for objective characterizations of pain and the potential applications of these methods to scale up the results of this work.

### Caveats and Limitations

Despite the clean new results and the congruence with the prior work based on the same data set, we caution that the modest size of the cohort limits our conclusions. The treatment of each participant as a random process guarantees the statistical power of each empirically estimated signature with 95% confidence interval. We ensured that the 4s-window with 50% overlap provided a continuous estimation with renewal of activity every 2 s comprising enough fluctuations to make a sound stochastic estimate and shift to the next point along the stochastic trajectory. However, the *n* of 21 participants, 9 with SOR is modest. We need a larger cohort. Further, the group was not balanced in sex and age. Ideally, we would like to sample larger numbers of males and females, but also examine transgender groups and groups with same-sex orientation. Lastly, it would be great to sample from other disorders of the nervous systems that also complain about issues with pain and temperature dysregulation.

## 5. Conclusions

Using this new approach, it will be possible to scale up our results from this modest cohort and ascertain subtypes of pain sensation. A positive note is that by integrating the complementary subjective and objective methods that we used here, we will attain much more than using only one method on its own right. In this sense, despite the caveats, we feel confident that the present methods have the potential to help us advance our understanding of the perception of physically induced pain—as registered by micro fluctuations in EEG brain signals.

## Figures and Tables

**Figure 1 jpm-11-00093-f001:**
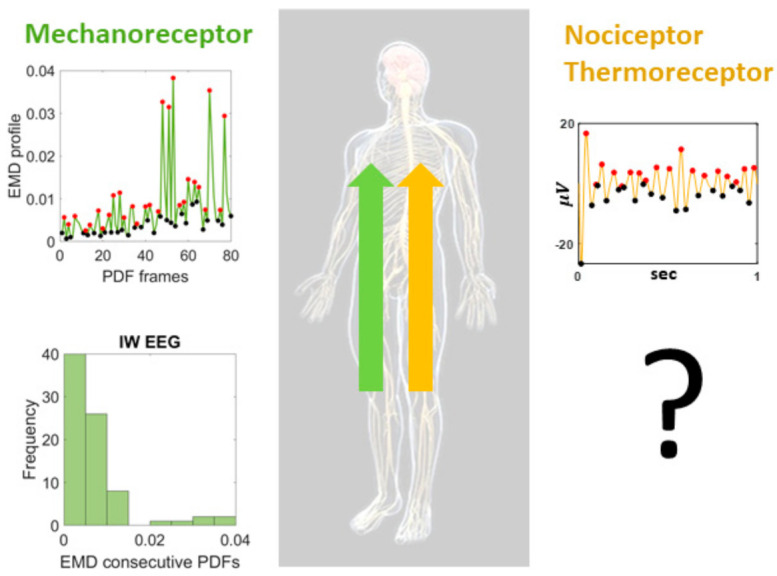
Proposed central characterization of lingering (pain) sensation. Special participant Ian Waterman (IW) lost his kinesthetic reafference but retained the sensations of pain and temperature. His electro-encephalographic (EEG) waveforms at rest, provide information about the shifts in probability density functions characterizing the distributions of fluctuations in peak activities in the lead electrodes with maximal clustering coefficient derived from the network of leads. Such shifts distribute exponentially, signaling a memoryless, random distribution of these activities, such that past events do not contribute to the prediction of future events. This type of distribution of his central EEG activities is congruent with the distribution of his movement-kinesthetic reafferent peripheral activities. What type of distributions could we find in individuals with intact kinesthetic reafference but sensory over-responsivity resulting in lingering sensations of temperature-induced pain?

**Figure 2 jpm-11-00093-f002:**
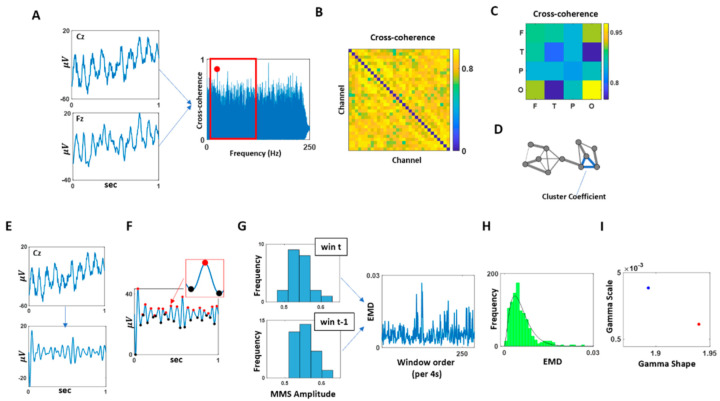
Data analytics pipeline. (**A**) For each pair of EEG channel combination, cross-coherence was computed, and its maximal value within the beta and gamma band (13–100 Hz) was extracted. (**B**) The maximal cross-coherence values obtained from (**A**) were used to construct an adjacency matrix of all EEG channel combinations. (**C**) EEG channel combinations were categorized by a combination of different scalp areas (F: frontal, T: temporal, P: parietal, O: occipital), and these categories’ median of maximal cross-coherence values, as shown in (**B**), were computed and compared. (**D**) Based on the channel’s adjacency matrix shown in (**B**), a network was constructed, where the nodes corresponded to each EEG channel, and the links corresponded to the maximal coherence values. As a measure of segregation of this network, cluster coefficients were computed and compared. (**E**) The channel with the highest cluster coefficient, computed at (**D**), was selected and its EEG waveform was band-pass filtered at 13–100 Hz. (**F**) The band-passed waveform was shifted up so that all values were positive. Then the spikes (maxima; denoted in red) and valleys (minima; denoted in black) were extracted to compute MMS (micro-movement spikes; standardized spike amplitudes), where the MMS is computed as dividing the spike value by the sum of the spike value and the average of the signal values between the two local minima as shown in Equation (2). (**G**) For each 4 s time window, MMS were gathered and plotted a histogram. For two consecutive time windows, the earth mover’s distance (EMD) was computed and compiled across time for each condition (5 min duration). (**H**) Histogram of EMDs were plotted and fitted to a Gamma PDF. (**I**) The fitted Gamma parameters obtained at (**H**) were plotted on the Gamma parameter plane, and its parameters were compared across conditions and groups.

**Figure 3 jpm-11-00093-f003:**
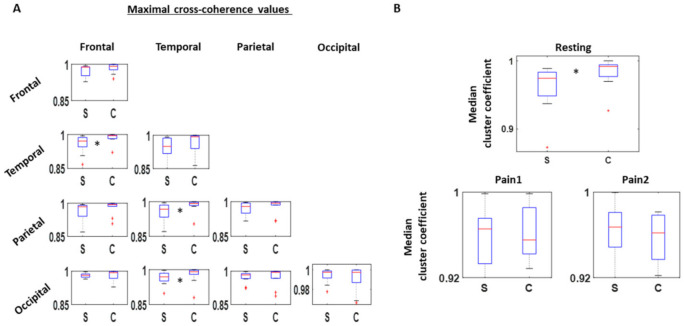
Cross-coherence between EEG channels. (**A**) 32 channels are categorized to one of these areas—Frontal, Temporal, Parietal, and Occipital—and maximal cross-coherence value examined within the beta and gamma band (13–100 Hz) for all channel pairs. Median values among the different channel pairs categorized by the scalp areas are then compared between SOR (S) and control (C) groups, where SOR exhibits lower values than the controls, particularly between frontal and temporal channels, parietal and temporal channels, and occipital and temporal channels. This pattern is found only during the resting condition. (**B**) Based on the adjacency matrix of maximal cross-coherence values, cluster coefficients are computed for all channels. The median of cluster coefficients is compared between the SOR and control groups for all three conditions. Cluster coefficient values are lower for SOR than the control group during the resting condition, but not significantly different when pain is induced.

**Figure 4 jpm-11-00093-f004:**
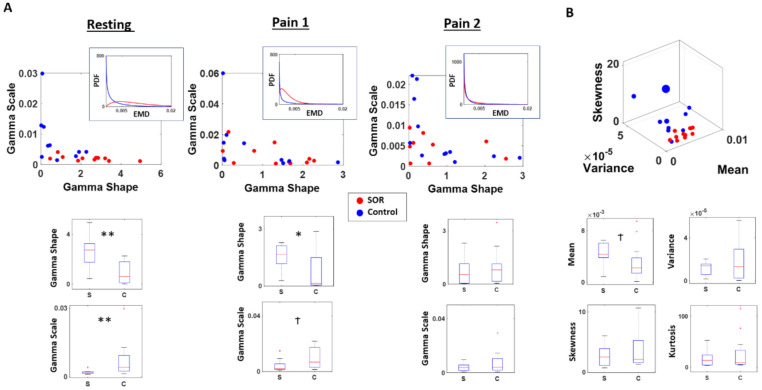
Differential localization in probability space and faster rate of change across pain conditions in SOR than controls. Stochastic shifts across time characterized by the distribution of EMDs between sequential time windows of MMS distributions reveal the departure of SOR from controls. (**A**) Distribution of EMDs obtained from sequential sliding windows of 4 s were examined and fitted a Gamma distribution. Under the resting state, SOR group tended to show a more symmetric (higher shape; more predictable) and less variable (lower scale; lower NSR) shifts in its EMD distributions. Under the first pain condition (Pain 1), the SOR group shifted distribution to a different regime tending to show less difference with the control group. Their PDF shifted to a less symmetric and less variable distribution; while the control group shifted to a lesser degree and mostly maintained its exponential distribution. Under the second pain condition (Pain 2), the SOR group showed even less difference with the control group, by exhibiting a more asymmetric (lower shape; more random) and more variable (higher scale; higher noise) pattern. (**B**) The distinction between the two groups can also be observed from the moments of EMD distributions, where the SOR tends to have a higher mean and a tighter range of variance and skewness than the control group. ** *p* < 0.01, * *p* < 0.05, Ϯ *p* < 0.10.

**Figure 5 jpm-11-00093-f005:**
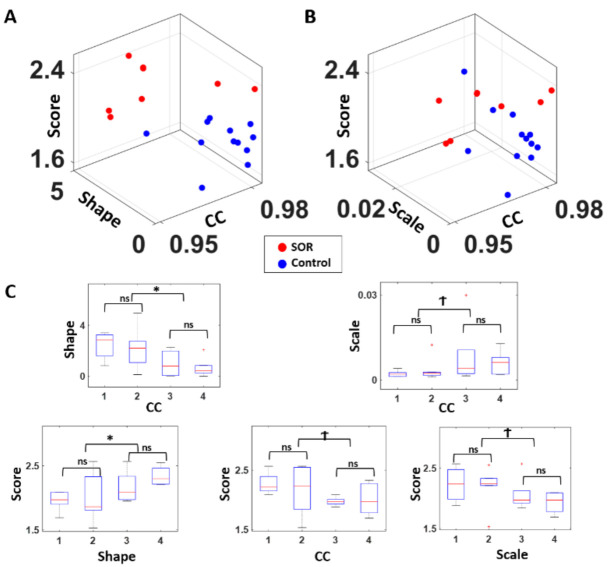
Congruence of clinical scores and stochastic signatures expressed in a parameter space spanned by score range, and stochastic signatures in the temporal, and frequency domains. (**A**) For each participant, the SRQ-IS was plotted on the *z*-axis (clinical score) along with the EMD’s fitted shape parameter on the *y*-axis (temporal) and the median cluster coefficient (CC) value of cross-coherence networks on the *x*-axis (frequency). Combining these metrics across 3 domains shows a good separation between the two groups. (**B**) A similar plot was made as (**A**), but with the EMD’s fitted scale parameter on the *y*-axis. Although the two groups show some separation, this visualization distinguishes the two groups slightly less than in (**A**), where the fitted Gamma shape parameter was utilized. (**C**) For statistical comparison, all participants were median ranked by cluster coefficients (CC; ranked in descending order) and expressed relative to the shape ((**C**) top-left) and scale ((**C**) top-right), with statistically significant differences between the extreme ranked quartiles in the shape parameter. The EMD’s fitted shape and scale parameters and cluster coefficients ((**C**) bottom subpanels) were also categorized into 4 ordered-ranked groups, and the SQR-IS (score) were compared between the upper and lower 50 percentiles; and between the lowest quartile and 2nd lowest quartile; and between the highest quartile and the 2nd highest quartile. Noticeably, all metrics show statistically significant correspondence between each other at a coarse level (as the upper and lower 50 percentiles show differences) but do not correspond at a finer level (as shown by the similarity between the 1st and 2nd quartiles, and 3rd and 4th quartiles). * *p* < 0.05, Ϯ *p* < 0.10.

## Data Availability

The data presented in this study are available on request from the corresponding author. The data are not publicly available due to health privacy issues.

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
