# Peer review of "Personalized Biometrics of Physical Pain Agree with Psychophysics by Participants with Sensory over Responsivity"

_jpm, 2021, doi:10.3390/jpm11020093_

Round 1

Reviewer 1 Report

The current study aims to examine the relationship between biometrics of physical pain and psychological perception of pain in participants with sensory over responsivity. In general, this study is interesting and adds to the literature. However, methods and results are difficult to interpret in the absence of clearly stated aims and hypotheses. In addition, authors did not lay out a clear justification for the inclusion and comparison of biometrics with psychological perception based on the introduction. Additional comments by section are provided below:

Introduction:

-authors are encouraged to flesh out their introduction of the psychological experience of pain. Being one of the central outcomes of this study, their discussion in this section is way too simplistic.

-the final paragraph is confusing and does not adequately lay the foundation for study aims and design (whether exploratory or otherwise).

Methods:

-How were psychological disorders ruled out? Also, what is a psychosocial disorder? Not familiar with this terminology and I venture this may not have objective diagnostic characteristics.

-Participants were asked to “have sufficient sleep the night before.” How was this tracked? Self-report?

-How were participants assigned to one of the three conditions? Random? While this may have been outlined in previous publications, a brief mention of this is warranted.

-No mention in this section of how “psychological perception” was measured or even operationalized for that matter...

Discussion:

-Authors mention in the discussion for the first time that their use of the term “psychological” refers to the “mental” perception of pain. There is no clear established precedent to use these terms synonymously and I strongly recommend that authors reconsider the use of "psychological" in their intended context given its strong connection with emotional aspects of the pain experience (e.g., biopsychosocial model of pain) rather than basic mental processes.

Author Response

The current study aims to examine the relationship between biometrics of physical pain and psychological perception of pain in participants with sensory over responsivity. In general, this study is interesting and adds to the literature.

Response: We thank the reviewer for the kind words and proceed to address the points raised.

However, methods and results are difficult to interpret in the absence of clearly stated aims and hypotheses. In addition, authors did not lay out a clear justification for the inclusion and comparison of biometrics with psychological perception based on the introduction. Additional comments by section are provided below:

Response: This work uses a data-driven approach, whereby we start without a clear hypothesis and let the data drive us to a set of testable propositions from the statistical patterns that empirically self-emerge from the data.

We agree with the reviewer that we did a poor job at stating the hypothesis once we see these patterns, as we were very conservative in our inferences and interpretation of these results. We have now stated several possible avenues to further explore various hypotheses in the Discussion section of the MS. Furthermore, we state in the Methods and Analyses the data -driven nature of our work, to distinguish it from the more traditional a priori significant hypothesis testing approach that we did not follow.

Introduction:

-authors are encouraged to flesh out their introduction of the psychological experience of pain. Being one of the central outcomes of this study, their discussion in this section is way too simplistic.

Response: This is now elaborated in the Introduction please see lines 47-53. 

-the final paragraph is confusing and does not adequately lay the foundation for study aims and design (whether exploratory or otherwise).

Response: We have now specified that this is a data driven approach on lines 118-120, and throughout the paragraph we explain the use of gross data (i.e., empirically estimate and analyze all fluctuations rather than smoothing out most fluctuations by a priori assuming a mean from a theoretical parametric model) We distinguish our approach from the traditional hypothesis testing paradigm.

Last paragraph in the Introduction reads:

The type of analysis that we offer here, away from assumptions of theoretical population statistics, has been previously used on a characterization of stochastic variations in movement reafferent signals. This is a data-driven approach whereby we let the data reveal patterns and then, upon interpretation and inference, we propose possible lines of inquiry to pursue in future work. In our prior work, the results led to automatic clustering of the above-mentioned clinical disorders on a spectrum [39]. These in turn have shown strong ties with other disorders of the nervous systems and various types of disruption in reafferent flow of movement information [40, 41], thus allowing us to further pursue new lines of questions. Since pain and temperature share separable afferent channels from movement afference, and crosstalk can be quantified through central processing using controlled experimental assays, here we apply these new data-driven analytical methods to SOR participants who suffer from abnormally high pain sensation. We re-examine previously published EEG data [34] as well as explore pain-evoked EEG responses induced by sustained temperature in controls vs. SOR participants. We do so by analyzing the gross data commonly discarded as noise, by avoiding a-priori assumptions of theoretical normal distributions of the fluctuations in EEG-waveforms’ peak amplitudes. We discuss our results within the context of stochastic processes amenable to offer a probabilistic account of pain sensation in general.

Methods:

-How were psychological disorders ruled out? Also, what is a psychosocial disorder? Not familiar with this terminology and I venture this may not have objective diagnostic characteristics.

Response: Psychological disorders are mental illnesses, which are usually diagnosed though clinical inventories with unifying (subjective) criteria. In this case, SOR is not considered a mental illness as e.g., schizophrenia, autism, ADHD, etc. in the Diagnostic Statistical Manual (in the US by the American Psychiatric Association) or in the International Psychiatric Classification, ICD-11 by the World Health Organization (WHO). They were interviewed using a medical survey, to rule out Psychological disorders. This is stated on page 4 line 147 (please, see below)

The study included 21 healthy participants (5 males and 16 females) between the ages 18 and 40 years old recruited from a convenience sample in a laboratory database. Participants were naïve to the testing. Based on a medical survey, participants with no chronic pain history and no regular use of analgesic or psychiatric medication were included in the study. Participants with any psychological, psychosocial, metabolic, and neurological disorders were excluded from the study. Participants were able to communicate and understand the instructions of the study. They self-reported to be free of any pain relief medications 24 hr prior and any caffeine products at least 2 hr prior to the experiment, and to have had sufficient sleep the night before. All participants provided written consent, which was approved by the Institutional Review Board of Rambam Health Care Campus (Haifa, Israel).

-Participants were asked to “have sufficient sleep the night before.” How was this tracked? Self-report?

Response: This was self-reported. We remark this point in the Methods section line 150-151.

-How were participants assigned to one of the three conditions? Random? While this may have been outlined in previous publications, a brief mention of this is warranted.

Response: All participants were assigned to undergo all three conditions. We clarified this in the Methods section line 167-172.

-No mention in this section of how “psychological perception” was measured or even operationalized for that matter...

Response: Psychological perception was operationally measured by the numerical rating scale. We clarified this in the Methods section line 180-181. We have also dropped the work psychological from psychological perception and used only perception throughout the paper, to avoid confusion.

Discussion:

-Authors mention in the discussion for the first time that their use of the term “psychological” refers to the “mental” perception of pain. There is no clear established precedent to use these terms synonymously and I strongly recommend that authors reconsider the use of "psychological" in their intended context given its strong connection with emotional aspects of the pain experience (e.g., biopsychosocial model of pain) rather than basic mental processes.

Response: We have used the term perception of pain throughout the paper, to avoid confusion.

Reviewer 2 Report

This is a well-written, interesting paper that takes an initial (feasibility-stage) look at characterizing pain in participants with Sensory Over Responsitivity (SOR) by analyzing their EEG data. The overall methodology/approach looks good, but I think that the discussion of the results are a bit lacking/incomplete.

Specifically, Fig 4A shows a trajectory of how the distribution of EMD values changes across the different conditions, and it is discussed in the paper that the SOR group exhibits a more symmetric pdf which is indicative of better 'predictability' than the exponential-like pdf. It is not clear what exactly can be predicted here. Can this be clarified? It also appears that the distribution for the SOR group from resting -> pain 1 -> pain 2 tends towards the memory-less, less-predictable, exponential distribution, and this is noted in the text, but there is no clear subsequent discussion on why the EMD distribution might have shifted in this fashion from resting -> pain 1 -> pain 2 -- it would be really interesting to see what the authors think about this trajectory.  Why does 'pain 2' essentially look the same for the control and SOR groups? What are the implications of the shift from more to less symmetric shapes, to exponential, in the SOR group?  The fact that there is this shift shown in the results is stated on lines 458-460, but this statement is not clearly followed by a hypothesis as to the possible reasons behind this shift -- only that this shift exists.  I think the observations of the results are reasonably well stated in this paper, but the actual discussion regarding these observations could be improved upon.

Another area that can be expanded on to improve the paper is more detail regarding the channels that had the highest cluster coefficients across patients. It would be interesting to see which channels were actually selected across patients, and where they are (spatially) - did they all cluster in the same general area?  Or are the physical locations of these channels fairly random?  There could be a plot showing the locations of these selected channels + some discussion on this.

Other comments:

  • On line 461, it is stated that "we found good correspondence between the clinical classification scores and the stochastic signatures...".  What does this mean, exactly?  What is 'good' correspondence?  Can you elaborate on this?
  • Are 'pain 1' and 'pain 2' application conditions exactly the same, the only difference being that 'pain 2' follows 'pain 1'?  I wasn't sure if this was the case (wasn't explicitly clear in the paper).
  • It is noted that experiments were also run testing different bandwidths, like the beta and the gamma band separately, but no pattern was found in this configuration. Can some text be added to provide a possible reason/hypothesis for why only both bands combined actually produced results?  (beyond just stating that this was an empirical finding...?)
  • Figures 1, 2, 4, 5 plots contain some text that is too small and difficult to read.  Please make all text contained in figures legible, or remove them entirely if it is not necessary to read that text.
  • Figure 3A,B: It would be better if all these plots showed the same scale on the y-axis for easier visual comparison (e.g., all with a lower bound of 0.88 or something), maybe with the exception of the Occipital/Occipital plot.
  • Please check for typos (e.g., line 170: "C.During.." is missing a space after the period). I also assume that the formatting of the manuscript will be cleaned up prior to publication (e.g., lines 15-16, it looks like the document template text is still left in the paper: "Tel.: (optional; include country code; if there are multiple 15 corresponding authors, add author initials)").

Author Response

This is a well-written, interesting paper that takes an initial (feasibility-stage) look at characterizing pain in participants with Sensory Over Responsitivity (SOR) by analyzing their EEG data. The overall methodology/approach looks good, but I think that the discussion of the results are a bit lacking/incomplete.

Response: We thank the reviewer for the kind remarks. We also address the important points that the reviewer raises below.

Specifically, Fig 4A shows a trajectory of how the distribution of EMD values changes across the different conditions, and it is discussed in the paper that the SOR group exhibits a more symmetric pdf which is indicative of better 'predictability' than the exponential-like pdf. It is not clear what exactly can be predicted here.

Response: The word prediction refers to the fluctuations in signal amplitude and their accumulation towards a particular type of probability distribution. At the systems’ level, the micro-movements fluctuations derived from the neural (EEG) related activity provide a type of moment-to-moment information that can fall in two extremes: (1) the exponential distribution of the continuous Gamma family of distributions best fitting the frequency histogram of the spikes representing the fluctuations in activity at the left limiting end, when the shape parameter is 1 and the memoryless exponential distribution is present. (2) at the right limiting end, the Gaussian distribution provides another type of distribution fitting the empirical spike data. Between these two limiting types are skewed distributions with various degrees of dispersion and heavy tails, all of which are well fit by the continuous Gamma family in an MLE sense. We show this in various publications. A recent one here in panel C of the below figure published in https://www.mdpi.com/1424-8220/18/9/3117

We explored these distributions with regards to the limiting cases and found that initially the exponential memoryless distribution fit the data well, but as the sensation of pain is reportedly experienced, we quantify the shift towards the other limiting (Gaussian) end. We know from stochastic processes that the Gaussian represent states that are more predictive (future events can be reliably predicted by past events, with high probability.) These distribution shapes are away from the random case of the memoryless exponential with Gamma shape value 1. Furthermore, we have empirically characterized other biorhythmic data registered from the nervous systems and by now have an empirical characterization of the shifts in other clinical cases. We know that e.g., when there is an absence of afferent feedback from movements (in the case of Ian Waterman) the distribution of the micro movement spikes derived from EEG signals using this analytical pipeline, is exponential. We also know this presence of the exponential distribution in autism, where there is also a problem feeling their bodily motions, in Parkinson’s disease, where we can provide a form of noise cancellation based on the noise to signal ratio that we characterize with the exponential family too; in schizophrenia where the micro-movement spikes of their bodily kinematics also show these abnormalities in prediction for deliberate motions; among other empirical clinical cases whereby we have characterized the biorhythms as we did here.

This body of empirical data enables us to provide an interpretation of these processes in view of how close (or far) they are from the random memoryless exponential case, or from the opposite Gaussian case. Both cases provide a prediction of future events based on past events. In the exponential case (memoryless) past events do not contribute to the prediction of future events. Thus, the fluctuations from past signals do not provide information to predict the fluctuations of future signals. On the contrary, in the case away from the exponential and towards the Gaussian, past events contribute to the probability of predicting future events (fluctuations in the spike signal.) It is in this sense that the word prediction is used. In recent years we have extended the results to causal prediction using Granger causality where we also estimate lags, but those recent results are beyond the scope of the present work.

Can this be clarified? It also appears that the distribution for the SOR group from resting -> pain 1 -> pain 2 tends towards the memory-less, less-predictable, exponential distribution, and this is noted in the text, but there is no clear subsequent discussion on why the EMD distribution might have shifted in this fashion from resting -> pain 1 -> pain 2 -- it would be really interesting to see what the authors think about this trajectory.  Why does 'pain 2' essentially look the same for the control and SOR groups?

Response: Our interpretation of this shift back to the exponential is that the models that we are using reflect the adaptive nature of the system. In the resting à Pain1 case, there is a large change in the SOR that does not occur in the controls. We interpret this as the correlate of the lingering sensation of pain. This can be thought of as a buffering mechanism sustaining the activity level in a more predictable way, whereby the mean and variance are sustained moment by moment for that time window NSR (i.e., variance/mean). The more symmetric shape tending to the Gaussian, concentrates the activity (peaks of the micro movement spikes) with a density that allows prediction in a probabilistic sense, i.e. telling us the probability that within this time window the mean activity will be X. As the system adapts to the lingering sensation of pain, the Pain1 à Pain 2 case, the activity returns to the exponential (random and memoryless) case whereby the person does not feel the lingering sensation of pain with the same intensity as it did in resting à Pain 1. The activity went back to a random memoryless state with no memory (buffering of the activity) thus not sustaining the lingering sensation as in the initial stages of the experiment. In this sense, the proposed stochastic-process interpretation of the pain sensation is to have these two opposing limiting states along a continuum (random memoryless vs. predictive) instantiated by the distribution of the signals and how they change from moment to moment. The EMD in this case provides information about the shifts of the frequency histograms representing probabilities derived from the signals’ fluctuations. Of course, this is a proposition and will need validation with larger N, but we express this caveat in the section referring to these issues.

These ideas are now in the Discussion on page 12 from line 521 - 535

What are the implications of the shift from more to less symmetric shapes, to exponential, in the SOR group?  The fact that there is this shift shown in the results is stated on lines 458-460, but this statement is not clearly followed by a hypothesis as to the possible reasons behind this shift -- only that this shift exists.  I think the observations of the results are reasonably well stated in this paper, but the actual discussion regarding these observations could be improved upon.

Response: We agree with the reviewer and have now included the above explanation in the Discussion on page 12.

Another area that can be expanded on to improve the paper is more detail regarding the channels that had the highest cluster coefficients across patients. It would be interesting to see which channels were actually selected across patients, and where they are (spatially) - did they all cluster in the same general area?  Or are the physical locations of these channels fairly random?  There could be a plot showing the locations of these selected channels + some discussion on this.

Response: Absolutely, we agree with the reviewer that this is an important aspect of the study which we neglected to mention. We now include a new Figure in the Appendix reflecting the distribution of leads with the highest clustering coefficients across the cohort. We note there that the SOR (red)  have a broader distribution of channels with central leads included but an absence of leads in the occipital area. This contrasts with control participants (blue). They have an absence of central leads and a presence of occipital lead. There is overlap otherwise with other leads and a broader distribution across frontal and medial leads in SOR than controls. A larger N may reveal a more interesting picture, but at least now we appreciate some individual differences that could help us design other studies.

Other comments:

  • On line 461, it is stated that "we found good correspondence between the clinical classification scores and the stochastic signatures...".  What does this mean, exactly?  What is 'good' correspondence?  Can you elaborate on this?

Response: This refers to the automatic clustering of the participants in correspondence with their clinical classification: those with SOR (red points in Figure 4) separated from those without SOR (blue) and the separation was maximal along the Gamma scale (NSR or dispersion signaling estimated Gamma var/ estimated Gamma mean) which we appreciate as well in Figure 4B of the Gamma moments parameter space. This correspondence emerges from the data variability rather than from our classification. It just happens to be on average in correspondence with the clinical scores. What was most interesting to us was the initial baseline difference in SOR patients (defined by their clinical scores) and their noticeable shift from resting to pain 1. The control’s remain in the exponential regime so their renewal process (as discussed above) remains memoryless (no lingering pain sensation as the SOR); then the SOR return to a memoryless state closer to the controls. As they adapt to the pain in Pain1 to Pain2 and remain similar to the controls’ state, this could teach us about a potential therapeutic value of using these methods as outcome measures of this reportedly lingering sensation in one case vs the other.  If they reportedly do not feel the same level of lingering sensation in Pain1 to Pain 2, and this corresponds to the case of exponential distribution (de-adaptation) the question is how could that state be sustained? Have we found a way to track the sensation of pain? Further studies will be required to address this question based on the participants self-reports in combination with these digital biometrics.

  • Are 'pain 1' and 'pain 2' application conditions exactly the same, the only difference being that 'pain 2' follows 'pain 1'?  I wasn't sure if this was the case (wasn't explicitly clear in the paper).

Response: Yes, they are the same. This is now noted in the methods in line 171-172.

  • It is noted that experiments were also run testing different bandwidths, like the beta and the gamma band separately, but no pattern was found in this configuration. Can some text be added to provide a possible reason/hypothesis for why only both bands combined actually produced results?  (beyond just stating that this was an empirical finding...?)

Response: This is a complete guess and may have nothing to do with reality, but our interpretation of this is that attentional states (well characterized by the gamma band in the literature) and afferent motion states (beta band used to remove motion artifacts) may together help amplify the internal sensations that the person experiences when modifying internal afferent flow of sensations at the periphery and bringing it the conscious attention, i.e., the conscious perception of the person. By themselves, the gamma band may just give us a hint of attentional states and the beta band a hint of afferent motion affecting the overall neural signal. However, in the presence of this sensation of pain induced by temperature (another afferent peripheral signal) the two bands together may reflect attentional level of awareness of internal sensations traveling in an afferent direction from the periphery to the brain. Alone by itself, each one of these seems to code for mental vs. bodily states; yet together they may code for the afferent feedback in the brain and provide a correlate of a form of interoception. (but this may be complete smoke, it is our best guess)

We notice it now in the discussion as a very speculative point.

  • Figures 1, 2, 4, 5 plots contain some text that is too small and difficult to read.  Please make all text contained in figures legible, or remove them entirely if it is not necessary to read that text.

Response: At this point, we deem the texts in the figures to be essential, and we created them as large as possible. Although they may be hard to see in physical paper, they are legible in the electronic format in higher resolution. We will take your remark into consideration and double check that the figures are legible at the end of the publication process. It has been our experience that sometimes the PDF conversion tends to distort fonts, so we will insist to the typesetting step that they provide links to higher res figures.

  • Figure 3A,B: It would be better if all these plots showed the same scale on the y-axis for easier visual comparison (e.g., all with a lower bound of 0.88 or something), maybe with the exception of the Occipital/Occipital plot.

Response: Thank you for the suggestion. We re-created the figure so that the subplots are on the same scale.

  • Please check for typos (e.g., line 170: "C.During.." is missing a space after the period). I also assume that the formatting of the manuscript will be cleaned up prior to publication (e.g., lines 15-16, it looks like the document template text is still left in the paper: "Tel.: (optional; include country code; if there are multiple 15 corresponding authors, add author initials)").

Response: Done

Round 2

Reviewer 1 Report

Authors have done much to address the concern surrounding the lack of a-priori hypotheses and the general use of the term "psychological" in their paper. However, there are still minor issues that are present surrounding presentation of the methods section.

In their response, authors addressed what a "psychological disorder" is when my question actually was what a "psychosocial" disorder (line 151) is and why this varying terminology is used? How are psychosocial issues pathologized (i.e., connected with the term "disorder")?

Authors are encouraged to be more specific as to which "medical survey" was used to rule-out participants with psychiatric/psychological disorders. Who performed this survey? How was this person(s) trained?

Regarding sleep, how was "sufficient sleep" quantified? Given that this is a physiological biometrics study, these things should be more objectively stated as sleep can potentially impact sensory responsivity. 

Author Response

Authors have done much to address the concern surrounding the lack of a-priori hypotheses and the general use of the term "psychological" in their paper. However, there are still minor issues that are present surrounding presentation of the methods section.

Response: We thank the reviewer for the kind words.

In their response, authors addressed what a "psychological disorder" is when my question actually was what a "psychosocial" disorder (line 151) is and why this varying terminology is used? How are psychosocial issues pathologized (i.e., connected with the term "disorder")?

Response: Psychosocial disorders are those defined by problems with social interactions. An example is Autism Spectrum Disorders. In the US, they are classified by the Diagnostic Statistical Manual (DSM-5) using two criteria: (1) problems with social interactions and communication; (2) ritualistic, restrictive, and repetitive behaviors. ASD is also detected using the Autism Diagnostic Observation Schedule (ADOS-2) with similar criteria as DSM-5, but with additional sub-scores that allow for a more detailed breakdown of scores reflecting various social and affect components of the behaviors that the social exchange between the clinician and patient evoke.

How are psychosocial issues pathologized (i.e., connected with the term "disorder")?

Response: There are accreditation programs that confer certifications to diagnose various populations. Through evidence-based criteria (observation based) data is gathered from thousands of participants across many studies. Clinicians define criteria consisting of common phenotypic characteristics and come to a consensus to detect the disorder in social interactions. In the US, two main bodies are involved in disorders like autism, for instance: The American Psychiatric Association and the American Psychological Association. Other psychosocial disorders are schizophrenia, bipolar disorder, depression, among others defined by the DSM criteria. For example, in autism and ADHD (attention deficit disorder), there are drug treatments prescribed by the Psychiatrists using the DSM and there are behavioral interventions prescribed by the ADOS criteria. In the US, the latter are covered by the tax payers and exist officially at the schools. From a very early age, children with a diagnosis of autistic disorder or autism spectrum disorder go on to receive such services (including behavioral modification treatments to make them conform to social norms dictated as well by these criteria.) These are issues beyond the scope of the paper, but worthwhile noticing as psychosocial disorders are very much a part of the US culture today. It is not clear how they are in other cultures. Yet judging by ICD criteria, in the US we would be much better off adopting a different model altogether.

The relevance of psychosocial disorders to SOR (e.g. autism, depression, schizophrenia, etc. defined in the DSM-5) is that these disorders may or may not be recognized as having physiological issues (e.g. dysregulated afferent information concerning pain, temperature and movement-kinesthetic related issues) giving rise to a distorted bodily sensation and consequently, atypical perception of bodily sensations. In the specific case of autism / ASD, the DSM criteria changed from version IV to version 5, to include sensory issues (broadly defined) and to add ADHD as a possible comorbid disorder (ADHD is also defined as a disorder in the DSM) The DSM is updated by the American Psychiatric Association, whereas the ADOS is part of a business called Western Psychological Services https://www.wpspublish.com/ catering to consumers in clinical psychology. It is updated by a small group of clinicians who own the rights and receive royalties for it (this can be found on the manual description and corresponding papers.)

In the EU there is a different manual: (from the web “The ICD is produced by a global health agency with a constitutional public health mission, while the DSM is produced by a single national professional association. WHO's primary focus for the mental and behavioral disorders classification is to help countries to reduce the disease burden of mental disorders.” More can be found here https://www.apa.org/monitor/2009/10/icd-dsm

Authors are encouraged to be more specific as to which "medical survey" was used to rule-out participants with psychiatric/psychological disorders. Who performed this survey? How was this person(s) trained?

Response: From line 159 to line 168, we explain the questionnaire provided to the participants. This is a standardized scale that can be administered by scientists with a PhD in the laboratory upon IRB approval. If the person does not have a diagnosis of a psychosocial disorder (e.g. ASD, Schizophrenia, Depression, etc., i.e., a disorder affecting how the person socially interacts with others as per the DSM-5, or the ICD-10) they are administered the standardized scale SQR.

We have now added on line 152 “This means that if the participant had a diagnosis of any of the above-mentioned disorders, they did not sign up, nor did they participate in the study.”

The SOR group was comprised of those whose Sensory Responsive Questionnaire Intensity Scale (SRQ-IS; [43])-Aversive score exceeded 2 standard deviations from its mean. The control group was comprised of those with scores within the 2 SD from the mean. Note, the SRQ-IS is designed to clinically identify those with sensory modulation disorder and is comprised of Hedonic scores and Aversive scores [44]. The Aversive scores that were used as a criterion to categorize groups involve answering intensity levels (on a scale 1-5) on scenarios such as “Being in dark/unlit surroundings bothers me”, and “Watching T.V. / computer in a well-lit room bothers me”. Further details of participant recruitment can be found in [45]. By using these scores, we operationalized each participant’s perception of sensory experience. 

Regarding sleep, how was "sufficient sleep" quantified? Given that this is a physiological biometrics study, these things should be more objectively stated as sleep can potentially impact sensory responsivity. 

Response: On line 158 we add “Sufficient sleep means that participants did not have any complaints about sleep disturbances. We did not measure their sleep.”

Sleep is an important aspect of anyone’s life. These days sleep can be tracked with wearables (e.g. the Apple watch, Fitbits, etc. ) and provide a better account of the person’s restless motions during the night and of the person’s overall resting status. But at the time of this study, such means were not available. We just followed the simpler question of whether they had sufficient sleep, to the extent that one feels tired and restless when one has not sufficient sleep. Otherwise, if one feels well rested and did not have sleep issues, it is assumed that the person had sufficient sleep.

In future versions of this study, we will measure sleep beyond assuming sufficient sleep by the person’s report of feeling well-rested and having had good sleep.
